# Association of tumor necrosis factor-α-308G/A polymorphism with the risk of obstructive sleep apnea: A meta-analysis of 14 case-control studies

**Zhenlian Zhang**[1,2�], **Dilihumaier Duolikun**[1�], **Tingting Dang**[3], **Yuanyuan Wang**[3], **Lijuan Ma**[1,2], **Xueyun Ma**[1,2], **Qiaoling Yao**[1,2]*

1 Department of Physiology, School of Basic Medical Sciences, Xinjiang Medical University, Urumqi, Xinjiang, China, 2 Xinjiang Key Laboratory of Molecular Biology for Endemic Diseases, Xinjiang Medical University, Urumqi, Xinjiang, China, 3 Third Clinical Medical College, Xinjiang Medical University, Urumqi, Xinjiang, China

උ These authors contributed equally to this work.
* 49165627@qq.com

## Abstract

Although numerous studies have suggested the association between TNF-α-308G/A polymorphism and susceptibility to obstructive sleep apnea (OSA), the results remained controversial and ambiguous. We performed the present meta-analysis to derive a more precise estimation. The PubMed, Embase, Cochrane library, Web of Science, China National Knowledge Infrastructure, Wanfang databases, and Weipu databases (until January 8, 2022) were accessed to retrieve relevant articles. Pooled odds ratios (ORs) with 95% confidence interval (95% CI) were calculated using the STATA statistical software.Totally, fourteen studies involving 2595 cases and 2579 controls were enrolled in this meta-analysis. Pooled results demonstrated significant association between TNF-α-308G/A polymorphism and OSA risk for the overall population (allele model:OR = 1.87 [1.47, 2.38] (n = 14), dominant model: OR = 1.88[1.48, 2.39] (n = 14), recessive model:OR = 2.83 [2.00, 4.00] (n = 11), homozygous model:OR = 3.30 [2.32, 4.68] (n = 11), and heterozygous model:OR = 1.67 [1.36, 2.06] (n = 14); $P<0.001$, respectively).Subgroup analysis showed that in both Caucasians and Asians, the A allele conferred increased risk to OSA compared to the G allele (Caucasians: OR = 1.40[1.03, 1.90] (n = 5), $P = 0.033$, Asians: OR = 2.30 [1.62, 3.26] (n = 9), $P< 0.001$). In subgroup analysis restricted to hospital-based individuals, significant association between TNF-α-308G/A polymorphism and OSA risk was identified under each genetic model. Whereas, in population-based individuals, increased risk of OSA were only found in homozygous model (OR = 2.19[1.23, 3.90] (n = 3), $P = 0.008$) and recessive model (OR = 1.77 [1.00, 3.13] (n = 3), $P = 0.048$). There was a substantial between-study heterogeneity ($I^2 = 69.10\%$) across studies which was explained by source of control participants ($P = 0.036$) by meta-regression. The results of leave-one-out meta-analysis and publication bias suggested the reliability and stability of our results.This meta-analysis suggested that TNF-α-308A allele may be a risk factor for the development of OSA. However, large scale,multi-center and well-designed case-control studies are needed in the future.

**Funding:** This study was supported by Natural Science Foundation of Xinjiang Uygur Autonomous Region (No.2021D01C278) received by ZZL. The funders had no role in study design, data collection and analysis, decision to publish, or preparation of the manuscript.

**Competing interests:** The authors have declared that no competing interests exist.

## Introduction

Obstructive sleep apnea (OSA) is a prevalent sleep-related breathing disorder resulting from the repetitive obstruction of the upper airway during sleep with prevalence ranging from 9% to 38% in the general population. The prevalence rises with advancing age, male sex, and high body mass index [1]. Published studies reported the associations between OSA and increased risk of cardiovascular disease, cognitive impairment, cancer, and metabolic syndrome [2–5]. Although continuous positive airway pressure (CPAP) is the first-line therapy for OSA, improving its compliance and identifying novel therapy strategies are still actively studied. Thus, OSA remains a public health challenge around the world, up till now, its etiology and pathogenesis are not fully understood.

Recently, the heritability of OSA and the indices of OSA severity, such as apnea hypopnea index (AHI), respiratory disturbance index (RDI), and oxygen desaturation index (ODI) were reported by Szily and coworkers [6]. Moreover, published studies have shown that OSA is more prevalent in Asia than in Europe. Ethnic differences in severity and treatment were also illustrated, indicating the potential role of genetic factors in imparting the variance of OSA [7–9]. Genetic associations with OSA, including the tumor necrosis factor (TNF)-α-308G/A polymorphism, have been studied extensively [10]. This polymorphism is characterized by adenine (A) substituted for guanine (G), which induces the expression of TNF-α [11, 12]. Remarkably, compared with control subjects, patients with OSA have higher systemic TNF-α levels, which showed a positive correlation with OSA severity [13]. Taken together, these findings suggest that TNF-α-308G/A polymorphism may be involved in the pathogenesis of OSA.

Although numerous studies have investigated the association between TNF-α-308G/A polymorphism and susceptibility to OSA, the results remained controversial and ambiguous [14, 15], which may be related to ethnicity, source of control, genotyping method, sample size, and so on. TNF-α-308G/A polymorphism in the promoter region of *TNF* gene is reported to be closely associated with OSA risk [14, 16], whereas others are not [15, 17]. Despite previous meta-analyses have been conducted [18–21], a number of relevant studies are not enrolled in their meta-analyses [14–16, 22, 23]. Hence, we performed a meta-analysis to derive a more accurate evaluation of the association between TNF-α-308G/A polymorphism and OSA susceptibility in all available studies. Moreover, source of the control participants was firstly used for subgroup analyses and confirmed to be the potential source of between-study heterogeneity by meta-regression.

## Materials and methods

### Search strategy

The PubMed, Embase, Cochrane library, Web of Science, China National Knowledge Infrastructure, Wanfang databases, and Weipu databases (until January 8, 2022) were used to perform a systematic search for the identification of studies addressing the association of TNF-α-308 G/A polymorphism and OSA. The adopted appropriate combinations of search terms were as follows: "obstructive sleep apnea syndrome", "OSAS", "obstructive sleep apnea", "OSA" and "tumor necrosis factor", "TNF", "tumor necrosis factor-a", "TNF-α" and "polymorphism", "gene", "variant", and "mutation". A manual search was also performed by reviewing the references of relevant studies and review articles.

### Inclusion and exclusion criteria

Studies that met the following criteria were included in the meta-analysis: (1) studies addressing the association of TNF-α-308 G/A polymorphism with OSA, (2) case–control design, (3)

adequate genotype distributions in cases and controls, and (4) full-text studies published in English and Chinese. The exclusion criteria we adopted were as follows: (1) enrolled patients without OSA; (2) enrolled patients without available genotype distribution; (3) letter, editorial, review or meta-analysis, abstract, conference papers, case report, studies on animals or cell-lines; and (4) duplicate studies. The definition of OSA was as described in the primary studies.

### Data extraction

The following data were extracted independently from each eligible study by two investigators (Zhang ZL and DD): first author's surname, year of publication, country, ethnicity, age, body mass index, gender (male/female), AHI of cases and controls, source of control, controls matched for, genotype method, Hardy–Weinberg equilibrium (HWE) in controls, sample size, and genotype frequencies in cases and controls. Discrepancies were settled through consultation. We contacted the authors by email if detailed data were missing.

### Quality assessment

The Newcastle–Ottawa Scale (NOS) captured the quality of eligible studies in terms of three parts, including "Selection", "Comparability", "Exposure", stratified into by low- (0–3), moderate- (4–6), and high-quality (7–9) levels in accordance with scores.

### Statistical analysis

The STATA statistical software (version 16.0) was used to carry out the meta-analyses. Pooled odds ratios (ORs) and 95% confidence intervals (CIs) were calculated using the Z test to evaluate the association of TNF-α-308 G/A polymorphism with OSA risk under the following genetic models: dominant (AA + AG vs. GG), recessive (AA vs. AG + GG), homozygous (AA vs. GG), heterozygote (AG vs. GG) and allele (A allele vs. G allele) models. The Pearson's $\chi 2$ test was performed to evaluate the HWE in control groups. The heterogeneity across studies was evaluated using the Cochran's Q and the $I^2$ statistics. Significant heterogeneity was considered when $P < 0.1$ for the Q-test or $I^2 > 50\%$. Thus, the random-effects model was applied to calculate the pooled ORs. Otherwise, the fixed-effects model was implemented. Subgroup analyses and meta-regression were conducted to explore potential sources of heterogeneity in total models according to ethnicity and source of control. The sensitivity analysis which was carried out with leave-one-out meta-analysis were performed for all genetic models to further confirm the reliability and stability of the results. Harbord funnel plot and Peters test were performed to access the publication bias for all models. $P$-value $< 0.05$ was considered to indicate statistical significance.

## Results

### Selection of studies

The process of study selection was presented in Fig 1. A total of 437 studies were initially identified through the database described above. 196 studies were screened after removing duplicates. By screening titles and abstracts, 148 studies were excluded because they were 1) meta-analyses or reviews ($n = 29$), 2) letters, editorials, abstracts, and case reports ($n = 10$), 3) studies on animals or cell lines ($n = 61$), 4) studies not on TNF-α ($n = 34$), studies not for case-control study($n = 1$), and 5) studies not on OSA ($n = 13$). A total of 48 full-text articles were further assessed for eligibility, and 34 articles were excluded because they were not for TNF-α rs1800629 ($n = 33$) or had insufficient data ($n = 1$) [24]. Finally, 14 studies were eligible for the

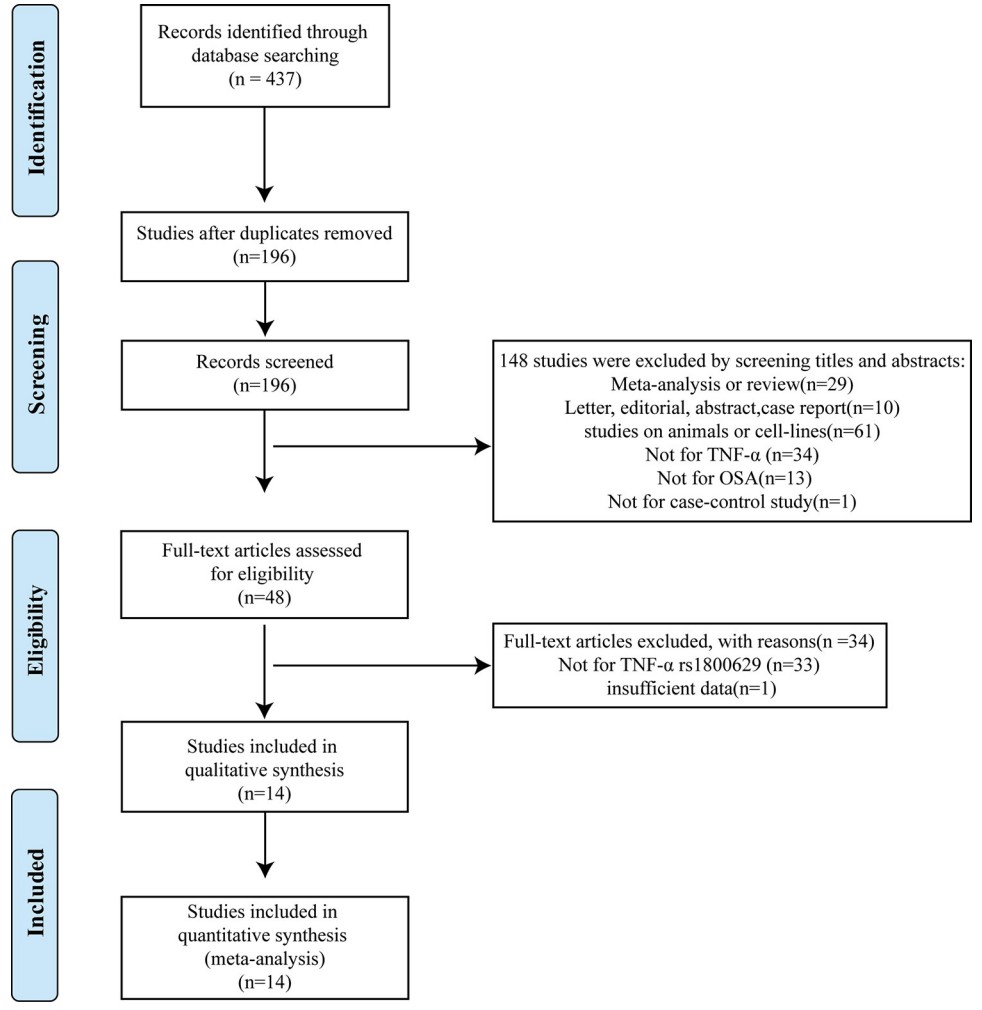

**Fig 1. Flow diagram of study selection (PRISMA format).**

inclusion criteria and enrolled in this meta-analysis with a total number of 5174 subjects consisting of 2595 cases and 2579 controls [14–17, 22, 23, 25–32].

## Characteristics of the included studies

As summarized in Table 1, all eligible studies were published from 2005 to 2021. In terms of study populations, nine studies were conducted in Asian and five in Caucasian. The sources of the control participants were hospital-based (HB) in nine studies and population-based (PB) in five studies. The polymerase chain reaction–restriction fragment length polymorphism (PCR-RFLP), TaqMan assay, DNA sequencing, KASP and Mass Array technology were used to identify the genotype in eligible studies. The HWE test was conducted for the control group in each eligible study. Only one study did not conform to the HWE law ($P < 0.05$) [26]. All eligible studies were of moderate or high quality according to the NOS quality assessment.

## Quantitative synthesis

Pooled results demonstrated significant association between TNF-α-308G/A polymorphism and OSA risk for overall population under specific genetic models (Fig 2A and Table 2) (allele

**Table 1. Basic characteristics of included studies.**

| author | year | case | control | case | | | control | | | HWE | source of control participants | genotype method | country | ethnicity | Quality score |
|---|---|---|---|---|---|---|---|---|---|---|---|---|---|---|---|
| | | | | GG | AG | AA | GG | AG | AA | | | | | | |
| Riha [32] | 2005 | 103 | 190 | 52 | 44 | 7 | 130 | 52 | 8 | 0.344 | PB | TaqMan | UK | Caucasian | 8 |
| Li [30] | 2006 | 24 | 48 | 11 | 10 | 3 | 35 | 12 | 1 | 0.981 | HB | TaqMan | China | Asian | 8 |
| Liu [31] | 2006 | 76 | 42 | 45 | 23 | 8 | 35 | 6 | 1 | 0.268 | HB | PCR-RFLP | China | Asian | 8 |
| Popko [28] | 2008 | 102 | 77 | 73 | 29 | 0 | 59 | 18 | 0 | 0.246 | PB | PCR-RFLP | Poland | Caucasian | 9 |
| Bhushan [26] | 2009 | 104 | 103 | 74 | 22 | 8 | 90 | 10 | 3 | 0.001 | HB | PCR-RFLP | Indians | Asian | 8 |
| Karkucak [27] | 2012 | 69 | 42 | 51 | 18 | 0 | 33 | 9 | 0 | 0.437 | HB | PCR-RFLP | Turkish | Caucasian | 8 |
| Almpanidou [25] | 2012 | 220 | 319 | 118 | 83 | 19 | 221 | 84 | 14 | 0.107 | PB | PCR-RFLP | Greek | Caucasian | 9 |
| Guan [29] | 2013 | 531 | 162 | 430 | 95 | 6 | 143 | 18 | 1 | 0.604 | HB | PCR-RFLP | China | Asian | 8 |
| Li [17] | 2013 | 155 | 100 | 137 | 18 | 0 | 88 | 12 | 0 | 0.523 | PB | DNA sequencing | China | Asian | 8 |
| Wang [22] | 2014 | 78 | 78 | 54 | 16 | 8 | 66 | 10 | 2 | 0.057 | HB | PCR-RFLP | China | Asian | 8 |
| Petrek [15] | 2015 | 116 | 351 | 85 | 30 | 1 | 248 | 98 | 5 | 0.175 | PB | Mass Array technology | Czech | Caucasian | 5 |
| Zhu [16] | 2015 | 200 | 200 | 124 | 46 | 30 | 166 | 31 | 3 | 0.278 | HB | DNA sequencing | China | Asian | 8 |
| Zhang [14] | 2019 | 750 | 800 | 576 | 153 | 21 | 656 | 132 | 12 | 0.077 | HB | TaqMan | China | Asian | 8 |
| Abulikemu [23] | 2021 | 67 | 67 | 42 | 20 | 5 | 60 | 6 | 1 | 0.098 | HB | KASP | China | Asian | 7 |

Abbreviation

HWE, Hardy-Weinberg equilibrium

PB,population-based

HB,hospital-based

PCR-RFLP:polymerase chain reaction-restriction fragment length polymorphism

KASP,kompetitive allele specific PCR.

model:OR = 1.87 [1.47, 2.38] (n = 14), dominant model: OR = 1.88[1.48, 2.39] (n = 14), recessive model:OR = 2.83 [2.00, 4.00] (n = 11), homozygous model:OR = 3.30 [2.32, 4.68] (n = 11), and heterozygous model:OR = 1.67[1.36, 2.06] (n = 14); $P<0.001$, respectively).Subgroup analysis according to ethnicity showed that in both Caucasians and Asians, the A allele conferred increased risk to OSA compared to the G allele(Fig 2B and Table 2) (Caucasians: OR = 1.40 [1.03, 1.90] (n = 5), $P = 0.033$, Asians: OR = 2.30 [1.62, 3.26] (n = 9), $P< 0.001$). In subgroup analysis restricted to HB individuals, significant association between TNF-α-308G/A polymorphism and OSA risk was identified under each genetic model (Fig 2C and Table 2). Whereas, in PB individuals, increased risk of OSA were only found in homozygous model (OR = 2.19

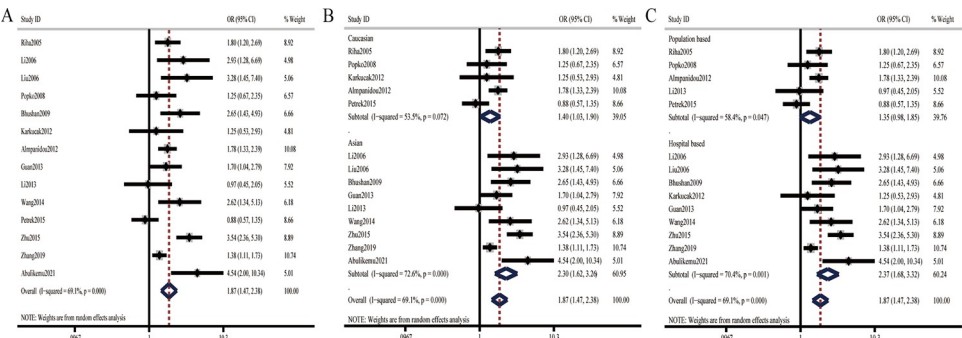

**Fig 2. Forest plots of the association between TNF-α-308G/A polymorphism and OSA under the allele model.** A: overall results; B: subgroup analysis by ethnicity; C: subgroup analysis by source of control.

**Table 2. Meta-analysis results of association between TNF-α-308G/A polymorphism and OSA.**

| Contrast model | | Subjects (cases/controls) | Studies,n | OR (95%CI) | $P^*$ | $I^2$(%) |
|---|---|---|---|---|---|---|
| Total studies | | | | | | |
| allele model | A vs. G | 5190/5158 | 14 | 1.87 (1.47, 2.38) | 0 | 69.10 |
| dominant model | AA/AG vs. GG | 2595/2579 | 14 | 1.88(1.48, 2.39) | 0 | 59.50 |
| recessive model | AA vs. AG/GG | 2269/2360 | 11 | 2.83 (2.00, 4.00) | 0 | 15.00 |
| homozygous model | AA vs. GG | 2269/2360 | 11 | 3.30 (2.32, 4.68) | 0 | 20.80 |
| heterozygous model | AG vs. GG | 2595/2579 | 14 | 1.67 (1.36, 2.06) | 0 | 38.80 |
| Subgroup analysis | | | | | | |
| Caucasians | | | | | | |
| allele model | A vs. G | 1220/1958 | 5 | 1.40(1.03, 1.90) | 0.033 | 53.50 |
| dominant model | AA/AG vs. GG | 610/979 | 5 | 1.48 (1.03, 2.13) | 0.035 | 56.30 |
| recessive model | AA vs. AG/GG | 439/860 | 3 | 1.77 (1.00, 3.13) | 0.048 | 0.00 |
| homozygous model | AA vs. GG | 439/860 | 3 | 2.19 (1.23, 3.90) | 0.008 | 0.00 |
| heterozygous model | AG vs. GG | 610/979 | 5 | 1.46 (1.04, 2.06) | 0.029 | 48.10 |
| Asians | | | | | | |
| allele model | A vs. G | 3970/3200 | 9 | 2.30 (1.62, 3.26) | 0 | 72.60 |
| dominant model | AA/AG vs. GG | 1985/1600 | 9 | 2.23 (1.59, 3.14) | 0 | 62.80 |
| recessive model | AA vs. AG/GG | 1830/1500 | 8 | 3.51 (2.06, 5.98) | 0 | 11.00 |
| homozygous model | AA vs. GG | 1830/1500 | 8 | 4.22 (2.32, 7.67) | 0 | 23.10 |
| heterozygous model | AG vs. GG | 1985/1600 | 9 | 1.86 (1.40, 2.47) | 0 | 39.60 |
| Subgroup analysis | | | | | | |
| HB | | | | | | |
| allele model | A vs. G | 3798/3084 | 9 | 2.37 (1.68, 3.32) | 0 | 70.40 |
| dominant model | AA/AG vs. GG | 1899/1542 | 9 | 2.30 (1.66, 3.20) | 0 | 58.70 |
| recessive model | AA vs. AG/GG | 1830/1500 | 8 | 3.51 (2.06, 5.98) | 0 | 11.00 |
| homozygous model | AA vs. GG | 1830/1500 | 8 | 4.22 (2.32, 7.67) | 0 | 23.10 |
| heterozygous model | AG vs. GG | 1899/1542 | 9 | 1.89 (1.45, 2.47) | 0 | 31.40 |
| PB | | | | | | |
| allele model | A vs. G | 1392/2074 | 5 | 1.35 (0.98, 1.85) | 0.068 | 58.40 |
| dominant model | AA/AG vs. GG | 696/1037 | 5 | 1.41 (0.97, 2.06) | 0.074 | 61.40 |
| recessive model | AA vs. AG/GG | 439/860 | 3 | 1.77 (1.00, 3.13) | 0.048 | 0.00 |
| homozygous model | AA vs. GG | 439/860 | 3 | 2.19 (1.23, 3.90) | 0.008 | 0.00 |
| heterozygous model | AG vs. GG | 696/1037 | 5 | 1.40 (0.98, 2.00) | 0.066 | 54.50 |

Abbreviation

HB: hospital-based

PB: population-based

Note:$P^*$:$P$ value for pooled OR.

[1.23, 3.90] (n = 3), $P$ = 0.008) and recessive model (OR = 1.77 [1.00, 3.13] (n = 3), $P$ = 0.048) (Table 2).

## Heterogeneity and sensitivity analysis

Significant between-study heterogeneity was considered in overall pooled analysis under allele, dominant and heterozygous models. Thus, random-effects model was applied to calculate the pooled ORs. Meta-regression was conducted to explore potential source of between-study heterogeneity, showing that the source of control participants caused the heterogeneity between studies in allele model ($P$ = 0.036) (Fig 3A). However, we did not find the potential source of

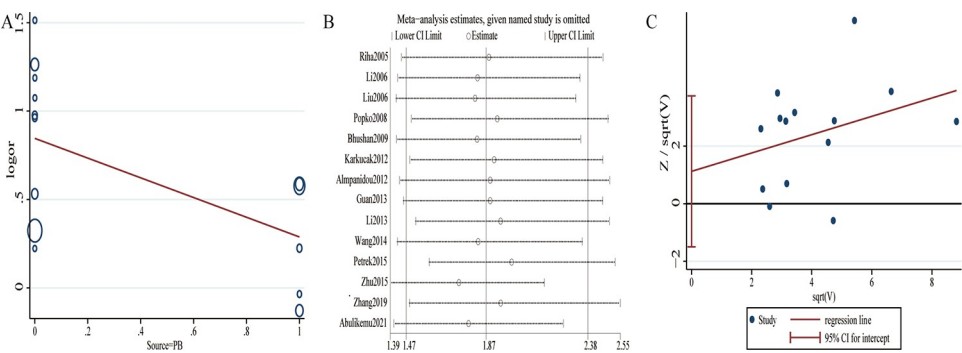

**Fig 3. Meta-regression, sensitivity analysis and publication bias in overall population under the allele model.** A: meta-regression($P$ = 0.036); B: sensitivity analysis; C: Harbord funnel plot (Harbord test $P$ = 0.368, Peters test $P$ = 0.120).

heterogeneity in dominant($P$ = 0.216 for ethnicity, $P$ = 0.124 for source of control participants) and heterozygous models($P$ = 0.303 for ethnicity, $P$ = 0.181 for source of control participants). Additionally, as presented in Fig 3B, results of leave-one-out meta-analysis showed no significant effect on the pooled ORs by any single study. Moreover, while omitting one study without confirmation of HWE [26], pooled ORs were not noticeably changed (OR = 1.83[1.42, 2.35]), further confirming the reliability and stability of present meta-analysis.

## Publication bias

Harbord funnel plot and Peters test were performed to estimate publication bias under all genetic models, and no evidence of publication bias was found in this meta-analysis (A vs. G: Harbord test $P$ = 0.368, Peters test $P$ = 0.120; AA/AG vs. GG: Harbord test $P$ = 0.212, Peters test $P$ = 0.097; AA vs. AG/GG: Harbord test $P$ = 0.810, Peters test $P$ = 0.300; AA vs. GG: Harbord test $P$ = 0.921, Peters test $P$ = 0.266; AG vs. GG: Harbord test $P$ = 0.131, Peters test $P$ = 0.084) (Fig 3C).

## Discussion

On the basis of 14 studies involving 2595 cases and 2579 controls, the present meta-analysis aimed to clarify the association between TNF-α-308G/A polymorphism and susceptibility of OSA. Pooled results demonstrated a significant association of TNF-α-308A allele and increased risk of OSA (OR = 1.87 [1.47, 2.38], $P$ < 0.001), indicating that TNF-α-308A allele may be a risk factor for the development of OSA.

Furthermore, results of subgroup analysis remarkably showed a 1.40-fold increased risk of OSA for Caucasians and a 2.30-fold increased risk of OSA for Asians with allele A, suggesting the significant influence of ethnicity on OSA risk. This finding is consistent under different genetic models and may explain the higher prevalence of OSA in Asians than in Europeans [9]. Moreover, under allele model, stratified analysis by source of control showed increased risk of OSA in HB studies (OR = 2.37 [1.68, 3.32], $P$ < 0.001) but not in PB studies ($P$ = 0.068), indicating that certain aspects of the study design may affect the results of a case–control study. Besides, results of the codominant model proved that carrying either genotype AA or genotype AG would put people at higher risk for OSA, which is consistent with the findings that level of TNF-α for individuals with genotypes AG and AA is higher than that of individuals with genotype GG [33].

Evidence showed that TNF inhibition was associated with decreased risk of OSA [34]. Interestingly, evidence demonstrated a high level of circulating TNF-α in patients with OSA, indicating that TNF-α may be a promising circulating biomarker to assess the degree of OSA [13, 35–37]. We have comprehensively analyzed the results of 14 studies through present meta-analysis and found increased risk of OSA in those with TNF-α-308A allele, which may be explained by elevated TNF-α expression via regulating the promoter transcriptional activity [12].Moreover,TNF-α gene lies in the highly polymorphic major histocompatibility complex (MHC) region. Apart from TNF-α-308 G/A polymorphism, TNF-α-238 G/A (rs361525), TNF-α-857G/A (rs1799724), and TNF-α-1031 T/C (rs1799964) are identified. Therefore, the linkage disequilibrium between alleles across the MHC may be involved in regulating the expression of TNF-α. In addition to gene-gene interactions, gene-environment interactions may also play a role in regulating TNF-α expression and thus participating in the pathogenesis of OSA.Thus, although our results showed that TNF-α-308G/A polymorphism was an important gene locus affecting risk of OSA, the mechanism is complex and should be further studied.

Previous meta-analyses were conducted by Varvarigou et al. in 2011 (three studies) [20], Huang et al. in 2012 (four studies) [21], Zhong et al. in 2014 (seven studies) [19], and Wu et al. in 2014 (ten studies) [18]. Similar to the above meta-analyses, our results demonstrated a significant association of TNF-α-308A allele and increased risk of OSA. However, our meta-analysis was on the basis of 14 case–control studies involving 2595 cases,while the largest sample size in previous meta-analyses was 10 studies with 1522 patients [18], providing more convincing and precise evidence that TNF-α-308G/A polymorphism increased the susceptibility to OSA. Statistical similar OSA risk were observed both in European (1.68-fold) and Asian population (2.02-fold) carrying A allele by Wu et al. [18], whereas higher risk of OSA for Asians with allele A(2.30-fold) than Caucasians(1.40-fold) was found in the present meta-analysis. Moreover, source of control participants was firstly used for subgroup analyses, showing that TNF-α-308A allele increases the risk of OSA in HB case–control studies but not in PB studies, suggesting the influence of study design on results. Additionally, we further performed meta-regression and found that source of control participants may be the potential source of heterogeneity

## Study limitations

Although the results of sensitivity analysis and publication bias suggested the stability of our results, some limitations should be considered. First, language bias resulted from limitation to publication in English and Chinese. Second, the confounding bias created by unavailable factors in most eligible studies like sex, age, body mass index, life style, AHI, comorbidities, other gene polymorphisms that may play important roles in the pathophysiology of OSA. Third, sampling bias resulted from enrolling a study unconforming to HWE law [26]. Fourth, there could be additional factors that have not been included in this review.

In summary, the present meta-analysis indicated significant association between allele A and increased susceptibility of OSA. However, large scale,multi-center and well-designed case-control studies are also needed in the future. Meanwhile, it is also of great significance to explore the mechanism of TNF-α-308 G/A polymorphism involved in OSA pathogenesis.

## Supporting information

**S1 Table. PRISMA 2020 checklist.**
(DOCX)

**S2 Table. Search strategy.**
(DOCX)

**S1 Fig. Forest plots of the association between TNF-α-308G/A polymorphism and OSA under the dominant model.** A: overall results; B: subgroup analysis by ethnicity; C: subgroup analysis by source of control.
(TIF)

**S2 Fig. Forest plots of the association between TNF-α-308G/A polymorphism and OSA under the recessive model.** A: overall results; B: subgroup analysis by ethnicity; C: subgroup analysis by source of control.
(TIF)

**S3 Fig. Forest plots of the association between TNF-α-308G/A polymorphism and OSA under the homozygous model.** A: overall results; B: subgroup analysis by ethnicity; C: subgroup analysis by source of control.
(TIF)

**S4 Fig. Forest plots of the association between TNF-α-308G/A polymorphism and OSA under the heterozygous model.** A: overall results; B: subgroup analysis by ethnicity; C: subgroup analysis by source of control.
(TIF)

## Acknowledgments

We appreciate the contribution of participants and researchers in the studies included in this review.

## Author Contributions

**Conceptualization:** Zhenlian Zhang, Qiaoling Yao.

**Data curation:** Zhenlian Zhang, Dilihumaier Duolikun.

**Formal analysis:** Zhenlian Zhang, Dilihumaier Duolikun, Tingting Dang.

**Funding acquisition:** Zhenlian Zhang.

**Methodology:** Zhenlian Zhang, Yuanyuan Wang.

**Software:** Lijuan Ma, Xueyun Ma.

**Supervision:** Zhenlian Zhang, Qiaoling Yao.

**Validation:** Zhenlian Zhang, Qiaoling Yao.

**Visualization:** Zhenlian Zhang, Dilihumaier Duolikun, Tingting Dang, Yuanyuan Wang, Lijuan Ma, Xueyun Ma.

**Writing – original draft:** Zhenlian Zhang.

**Writing – review & editing:** Lijuan Ma, Xueyun Ma, Qiaoling Yao.

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
