## [Decision Letter · Decision Letter 0]

9 Aug 2022

PONE-D-22-04764Association of Tumor Necrosis Factor-α-308G/A polymorphism with the risk of obstructive sleep apnea: An updated meta-analysis of 14 case-control studiesPLOS ONE

Dear Dr. Yao,

Thank you for submitting your manuscript to PLOS ONE. After careful consideration, we feel that it has merit but does not fully meet PLOS ONE’s publication criteria as it currently stands. Therefore, we invite you to submit a revised version of the manuscript that addresses the points raised during the review process.

Your manuscript has been assessed by one peer-reviewer, who suggested that the manuscript could be improved upon by better reporting of the study's limitations. Please note that we have only been able to secure a single reviewer to assess your manuscript. We are issuing a decision on your manuscript at this point to prevent further delays in the evaluation of your manuscript. Please be aware that the editor who handles your revised manuscript might find it necessary to invite additional reviewers to assess this work once the revised manuscript is submitted. 

We look forward to receiving your revised manuscript.

Kind regards,

Maria Elisabeth Johanna Zalm, Ph.D

Editorial Office

PLOS ONE

Journal Requirements:

“This study was supported by Natural Science Foundation of Xinjiang Uygur Autonomous Region (No.2021D01C278) received by ZZL. The funders had no role in study design, data collection and analysis, decision to publish, or preparation of the manuscript.”

5**. **Please include captions for your Supporting Information files at the end of your manuscript, and update any in-text citations to match accordingly. Please see our Supporting Information guidelines for more information: http://journals.plos.org/plosone/s/supporting-information.

Reviewers' comments:

Reviewer's Responses to Questions

**Comments to the Author**

1. Is the manuscript technically sound, and do the data support the conclusions?

Reviewer #1: Yes

2. Has the statistical analysis been performed appropriately and rigorously? 

Reviewer #1: Yes

3. Have the authors made all data underlying the findings in their manuscript fully available?

Reviewer #1: Yes

4. Is the manuscript presented in an intelligible fashion and written in standard English?

Reviewer #1: Yes

5. Review Comments to the Author

Reviewer #1: This manuscript is a meta-analysis of the association between Tumor Necrosis Factor-α-308G/A polymorphism and obstructive sleep apnea. The statistical analysis of 14 case-control studies selected for this review was conducted rigorously and this meta-analysis is appropriately based on the Prisma checklist. The introduction of this manuscript reported previous meta-analyses in this area and explained that they included additional case-control studies.

The conclusions are based on the findings obtained through adequate statistical analysis. However, the discussion should include some limitations related to the genetic association studies and the fact that the studies included in this meta-analysis did not report the effect of AIH (apnea, hypoapnea index), oxygen saturation levels, arousals, comorbidities, sex, age, and other factors that could play an important role in the pathophysiology of the obstructive sleep apnea.

The methodology is appropriate and rigorous. The inclusion and exclusion criteria are well formulated. The statistical analysis used in this meta-analysis included: Pearson’s χ2 test performed to evaluate the Hardy–Weinberg equilibrium (HWE), Cochran’s Q and the I2 statistics to evaluate heterogeneity, Z and OR test to evaluate genetic models, Harbord funnel plot and Peters test to identify publication bias. The results of these analyses are shown in tables and figures (forest plots, Harbord funnel plot) and show complete information about the association between Tumor Necrosis Factor-α-308G/A polymorphism and obstructive sleep apnea reported in case-control studies.

Data from each case-control study is described in the manuscript. Prisma checklist is supporting information. Complete and broad statistic results are shown. The references included the articles selected for this meta-analysis and previous meta-analyses among others.

6. PLOS authors have the option to publish the peer review history of their article (what does this mean?). If published, this will include your full peer review and any attached files.

Reviewer #1: **Yes: **Liliana Otero

---

## [Author Response · Author response to Decision Letter 0]

18 Sep 2022

Journal Requirements:

Response: Thank you for this comment. We have carefully revised the manuscript according to the PLOS ONE style templates.All the revised portions of the manuscript are now marked in red (except in the clean version of the revised manuscript, also submitted).

Response: Thank you for this comment. Participant consent is not applicable since the present meta-analysis was based on published data.

“This study was supported by Natural Science Foundation of Xinjiang Uygur Autonomous Region (No.2021D01C278) received by ZZL. The funders had no role in study design, data collection and analysis, decision to publish, or preparation of the manuscript.”

Response: Thank you for this comment. The amended Funding Statement marked in red has been within our cover letter accordingly. 

Response: Thank you for this comment. Since the present meta-analysis was based on published data, all relevant data are within the manuscript and its Supporting information files.The updated Data Availability statement marked in red has been within our cover letter accordingly. 

Response: Thank you for this comment. We have listed Supporting Information captions at the end of the manuscript in a section titled “Supporting information”. 

Reviewer #1: 

This manuscript is a meta-analysis of the association between Tumor Necrosis Factor-α-308G/A polymorphism and obstructive sleep apnea. The statistical analysis of 14 case-control studies selected for this review was conducted rigorously and this meta-analysis is appropriately based on the Prisma checklist. The introduction of this manuscript reported previous meta-analyses in this area and explained that they included additional case-control studies.

The conclusions are based on the findings obtained through adequate statistical analysis. However, the discussion should include some limitations related to the genetic association studies and the fact that the studies included in this meta-analysis did not report the effect of AIH (apnea, hypoapnea index), oxygen saturation levels, arousals, comorbidities, sex, age, and other factors that could play an important role in the pathophysiology of the obstructive sleep apnea.

The methodology is appropriate and rigorous. The inclusion and exclusion criteria are well formulated. The statistical analysis used in this meta-analysis included: Pearson’s χ2 test performed to evaluate the Hardy–Weinberg equilibrium (HWE), Cochran’s Q and the I2 statistics to evaluate heterogeneity, Z and OR test to evaluate genetic models, Harbord funnel plot and Peters test to identify publication bias. The results of these analyses are shown in tables and figures (forest plots, Harbord funnel plot) and show complete information about the association between Tumor Necrosis Factor-α-308G/A polymorphism and obstructive sleep apnea reported in case-control studies.

Data from each case-control study is described in the manuscript. Prisma checklist is supporting information. Complete and broad statistic results are shown. The references included the articles selected for this meta-analysis and previous meta-analyses among others.

Response: Thank you for your comment.According to your suggestion, we have revised the second limitation in discussion:The etiology of OSA is likely multifactorial, however, sex, age, body mass index, life style, AHI, comorbidities, other gene polymorphisms and other factors that may play important roles in the pathophysiology of OSA were not available in most eligible studies. Therefore gene–gene or gene–environment interactions were not evaluated in the present meta-analysis.Thus, the second limitation may be unadjusted pooled estimates by potential confounding factors.

Thank you and best regards.

Yours sincerely,

Qiao ling Yao 

Corresponding author:Qiao-ling Yao,PhD

Department of Physiology, School of Basic Medical Sciences, Xinjiang Medical University, Urumqi, Xinjiang, China.

Email: 49165627@ qq.com

---

## [Decision Letter · Decision Letter 1]

20 Feb 2023

PONE-D-22-04764R1Association of Tumor Necrosis Factor-α-308G/A polymorphism with the risk of obstructive sleep apnea: An updated meta-analysis of 14 case-control studiesPLOS ONE

Dear Dr. Yao,

Thank you for submitting your manuscript to PLOS ONE. After careful consideration, we feel that it has merit but does not fully meet PLOS ONE’s publication criteria as it currently stands. Therefore, we invite you to submit a revised version of the manuscript that addresses the points raised during the review process.

We look forward to receiving your revised manuscript.

Kind regards,

Cho Naing, MBBS, PhD, FRCP

Academic Editor

PLOS ONE

Journal Requirements:

Additional Editor Comments (if provided):

Association of Tumor Necrosis Factor-α-308G/A polymorphism with the risk of obstructive sleep apnea: An updated meta-analysis of 14 case-control studies

Abstract

Line # 31

OSA was used for the first time in this section; kindly spell it out.

Line # 35

Please replace the term ‘used” with “accessed”

As the authors describe how to estimate association in the current metanalysis, it should be "pooled odds ratio or summary odds ratio."

Line # 40-42; Line # 44-45; Line # 48-49

Please provide number of studies for the respective effect estimates.

Line # 49-50

“Substantial between-study heterogeneity” is preferable.

“ p value” of the I2 test should be deleted.

Line # 51

Better to describe as “Source of control participants”.

What sensitivity analysis is it for? Please provide specifically.

Sensitivity analysis for "source of control participants" was reported in the text.

In the abstract, I feel a lack of recommendation in the conclusions. Based on the study limitations that the authors identified during the current research, please suggest future well-designed case control studies, etc.

TEXT

Line # 87

Please omit the word "an updated". Publications from the past were not done by this team.

Search Strategy

Please provide “search strategy for PubMed and Embase in supplementary so that the audiences can replicate.

Line #101

Criteria # 1 is a doubtful. Did you select studies that showed only susceptible to OSA, but not protective. In earlier part, the authors claimed that “the results (from single studies) remained controversial and ambiguous”.

The authors should have selected studies that assessed association between TNF-α-308G/A polymorphisms and OSA (regardless of direction of association)

Line # 111

“The definition of OSA was as described in the primary studies” is preferable.

Line # 117

HWE should have been spelled out as the authors didn't use an abbreviation.

Line # 120

"Missing" is preferable than "ambiguous" as a description.

Line # 124

To the best of my understanding, the NOS checklist is a 9-star tool, with 9 being the highest possible star/score. It is impossible to score more than 9.

Line # 117

The abbreviation HWE may be used at this juncture (please refer to my comments on line #117).

Line # 155; 172-175; 178 Please provide number of studies for the respective effect estimates.

Line #188

This should come at the top of this paragraph.

Line # 198

For the dominant model and heterozygous models, please provide p values. What about the remaining models?

Please remove p value for I2 test.

Table #2

Please remove the last column with I2 test p value.

Discussion

This section requires more efforts.

As this is a meta-analysis, citations to individual studies should be reduced/shortened.

Please briefly describe how the current study and previous reviews in this field are compared and contrasted.

Please use a subheading of the “study limitations” for clarity

To add a recommendation as a concluding remark. Please note my recommendation for this section of the abstract.

Reviewers' comments:

Reviewer's Responses to Questions

**Comments to the Author**

1. If the authors have adequately addressed your comments raised in a previous round of review and you feel that this manuscript is now acceptable for publication, you may indicate that here to bypass the “Comments to the Author” section, enter your conflict of interest statement in the “Confidential to Editor” section, and submit your "Accept" recommendation.

Reviewer #2: (No Response)

2. Is the manuscript technically sound, and do the data support the conclusions?

Reviewer #2: Yes

3. Has the statistical analysis been performed appropriately and rigorously? 

Reviewer #2: Yes

4. Have the authors made all data underlying the findings in their manuscript fully available?

Reviewer #2: Yes

5. Is the manuscript presented in an intelligible fashion and written in standard English?

Reviewer #2: Yes

6. Review Comments to the Author

Reviewer #2: The manuscript was well written. Introduction clearly and adequately stated the background of the current issue of the obstructive sleep apnea. Nevertheless, there are some suggestions as follow.

Results:

Table 1- Basic Characteristics of included studies

In this table, suggest to provide reference number in addition to author names.

In figures, it would be great if authors could provide the forest plots of effect estimates for each genetic models as a supplemenatary file.

7. PLOS authors have the option to publish the peer review history of their article (what does this mean?). If published, this will include your full peer review and any attached files.

Reviewer #2: No

---

## [Author Response · Author response to Decision Letter 1]

5 Apr 2023

Dear Editor and Reviewers,

Thank you very much for all your insightful comments and suggestions. They have been very helpful. We carefully modified the manuscript accordingly. All the revised portions of the manuscript are now marked in red (except in the clean version of the revised manuscript, also submitted). Please, see below for our responses (in blue) to all the comments/suggestions.

Journal Requirements:

1.Please review your reference list to ensure that it is complete and correct. If you have cited papers that have been retracted, please include the rationale for doing so in the manuscript text, or remove these references and replace them with relevant current references. Any changes to the reference list should be mentioned in the rebuttal letter that accompanies your revised manuscript. If you need to cite a retracted article, indicate the article’s retracted status in the References list and also include a citation and full reference for the retraction notice.

Response: Thank you for this comment. We have carefully reviewed reference list and ensureed that it is complete and correct.

Additional Editor Comments:

Abstract

Line # 31

OSA was used for the first time in this section; kindly spell it out.

Response: Thank you for this comment. We have spelled it out.

Line # 35

Please replace the term ‘used” with “accessed”

As the authors describe how to estimate association in the current metanalysis, it should be "pooled odds ratio or summary odds ratio."

Response: Thank you for this comment. We have replaced the term “used” , “pooled odds ratio” with “accessed”and “pooled odds ratio”,respectively.

Line # 40-42; Line # 44-45; Line # 48-49

Please provide number of studies for the respective effect estimates.

Response: Thank you for this comment. We have provided number of studies for the respective effect estimates.

Line # 49-50

“Substantial between-study heterogeneity” is preferable.

“ p value” of the I2 test should be deleted.

Response: Thank you for this comment. We have replaced the term “statistically significant heterogeneity” with “Substantial between-study heterogeneity”. “ p value” of the I2 test has be deleted.

Line # 51

Better to describe as “Source of control participants”.

Response: Thank you for this comment. We have replaced the term “source of control” with “Source of control participants”.

What sensitivity analysis is it for? Please provide specifically. Sensitivity analysis for "source of control participants" was reported in the text.

Response: Thank you for this comment. Sensitivity analysis refers to the method of testing the robustness of the results obtained under certain assumptions, and observing whether the pooled results have changed before and after changing some important factors, so as to judge the robustness of the results of meta-analysis. In this study,we conducted a sensitivity analysis by removing individual studies one by one to observe the difference between the combined and total effects of the remaining studies, and examined the impact of individual studies. As described in line199-200: as presented in Fig 3B, results of sensitivity analysis showed no significant effect on the pooled ORs by any single study, further confirming the reliability and stability of the results.

In the abstract, I feel a lack of recommendation in the conclusions. Based on the study limitations that the authors identified during the current research, please suggest future well-designed case control studies, etc.

Response: Thank you for this comment. Recommendation has been added in the conclusions:However, large scale,multi-center and well-designed case-control studies are also needed in the future.

TEXT

Line # 87

Please omit the word "an updated". Publications from the past were not done by this team.

Response: Thank you for this comment. The word "an updated" has been omitted.

Search Strategy

Please provide “search strategy for PubMed and Embase in supplementary so that the audiences can replicate.

Response: Thank you for this comment. Search strategy has been provided in S2 table.

Line #101

Criteria # 1 is a doubtful. Did you select studies that showed only susceptible to OSA, but not protective. In earlier part, the authors claimed that “the results (from single studies) remained controversial and ambiguous”.

The authors should have selected studies that assessed association between TNF-α-308G/A polymorphisms and OSA (regardless of direction of association)

Response: Thank you for this comment.The word " the susceptibility of " has been omitted.

Line # 111

“The definition of OSA was as described in the primary studies” is preferable.

Response: Thank you for this comment. We have replaced the term “the definition of OSA used in each eligible study was accepted” with “The definition of OSA was as described in the primary studies”.

Line # 117

HWE should have been spelled out as the authors didn't use an abbreviation.

Response: Thank you for this comment. We have spelled it out.

Line # 120

"Missing" is preferable than "ambiguous" as a description.

Response: Thank you for this comment. We have replaced the term “ambiguous” with “Missing”.

Line # 124

To the best of my understanding, the NOS checklist is a 9-star tool, with 9 being the highest possible star/score. It is impossible to score more than 9.

Response: Thank you for this comment. We have changed it to 9.

Line # 117

The abbreviation HWE may be used at this juncture (please refer to my comments on line #117).

Response: Thank you for this comment. The abbreviation HWE has been used.

Line # 155; 172-175; 178 Please provide number of studies for the respective effect estimates.

Response: Thank you for this comment. We have provided number of studies for the respective effect estimates.

Line #188

This should come at the top of this paragraph.

Response: Thank you for this comment. It has been at the top of this paragraph

Line # 198

For the dominant model and heterozygous models, please provide p values. 

Response: Thank you for this comment. P values for the dominant model and heterozygous models have been provided. 

What about the remaining models?

Response: Thank you for this comment. Meta-regression was conducted to explore potential source of between-study heterogeneity, therefore, we only conducted meta-regression analysis on allele,dominant and heterozygous models.

Please remove p value for I2 test.

Response: Thank you for this comment. We did not remove the P value(P=0.036) because it is the P value of meta regression.

Table #2

Please remove the last column with I2 test p value.

Response: Thank you for this comment. We have removed the last column with I2 test p value.

Discussion

This section requires more efforts.

As this is a meta-analysis, citations to individual studies should be reduced/shortened.

Please briefly describe how the current study and previous reviews in this field are compared and contrasted.

Response: Thank you for this comment. We have carefully modified the discussion.

Please use a subheading of the “study limitations” for clarity

Response: Thank you for this comment. Subheading of the “study limitations” has been added.

To add a recommendation as a concluding remark. Please note my recommendation for this section of the abstract.

Response: Thank you for this comment.Recommendation has been added: however, large scale,multi-center and well-designed case-control studies are also needed in the future. Meanwhile, it is also of great significance to explore the mechanism of TNF-α-308 G/A polymorphism involved in OSA pathogenesis.

Reviewers' comments:

Review Comments to the Author

Reviewer #2: The manuscript was well written. Introduction clearly and adequately stated the background of the current issue of the obstructive sleep apnea. Nevertheless, there are some suggestions as follow.

Results:

Table 1- Basic Characteristics of included studies

In this table, suggest to provide reference number in addition to author names.

Response: Thank you for this comment. Reference number has been added in table1.

In figures, it would be great if authors could provide the forest plots of effect estimates for each genetic models as a supplemenatary file.

Response: Thank you for this comment. The forest plots of effect estimates for each genetic models have been provided as supplemenatary figures.

Thank you and best regards.

Yours sincerely,

Qiao ling Yao 

Corresponding author:Qiao-ling Yao,PhD

Department of Physiology, School of Basic Medical Sciences, Xinjiang Medical University, Urumqi, Xinjiang, China.

Email: 49165627@ qq.com

Mobile Number:15022917911

---

## [Editor Report · Decision Letter 2]

2 May 2023

PONE-D-22-04764R2Association of Tumor Necrosis Factor-α-308G/A polymorphism with the risk of obstructive sleep apnea: An updated meta-analysis of 14 case-control studiesPLOS ONE

Dear Dr. Yao,

Thank you for submitting your manuscript to PLOS ONE. After careful consideration, we feel that it has merit but does not fully meet PLOS ONE’s publication criteria as it currently stands. Therefore, we invite you to submit a revised version of the manuscript that addresses the points raised during the review process.

We look forward to receiving your revised manuscript.

Kind regards,

Cho Naing, MBBS, PhD, FRCP

Academic Editor

PLOS ONE

Journal Requirements:

Additional Editor Comments:

Association of Tumor Necrosis Factor-α-308G/A polymorphism with the risk of obstructive sleep apnea: An updated meta-analysis of 14 case-control studies"

The authors still need to revise an array of points. I also recommend reviewing the manuscript for language proficiency.

Title

Regarding my background comments, the phrase "an updated" has to be removed from the title.

Previous publications were not done by this team.

Abstract

Line #32………. “Obstructive” should not be capitalized

Line #41……… Please include the names of genetic models that were created for overall population. I believe that overall population was assessed using five different genetic models such as allele. "Dominant, recessive, homozygous, and heterozygous models"

Line # 47, 49……….. To spell out HB and PB

Line # 50…………. Name the genetic models for AA in comparison to GG, AG/GG

Line # 53…………. The statement "a leave-one-out meta-analysis" should be applied.

Line # 56……… Take out the word “also”

TEXT

Line # 63……….. replacing "published studies reported" for "increasing evidence has demonstrated" Line # 68/ 69………. Please rephrase this sentence. The word "thus" was used twice

Line # 72………. The wording "by Szily and coworkers or By Szily and associates" should have been used.

Line # 72………. Replace the word "strong evidence" with lighter tone, like "published studies."

Line # 78, 79, 80…… The authors could combine these two sentences into one and then cite [13] afterward.

Line # 90………. “Hence” is a preferrable term.

Line # 92………. “Source of the control participants”

Line # 114……… Please, remove “additionally”.

Line # 133……… Please keep "homozygous" and "heterozygous" while removing "codominant."

Line # 138……The word .”Otherwise” is preferrable to “By contrast”..

Line # 141……. So as to differentiate it from statistical models (Random. Fixed models), it needs to be mentioned as "for all genetic models".

Line # 141-142………… Are you referring to "a leave-one-out metanalysis" for the stability of the results.

Line # 146……… Literature search and characteristics of included studies.

Please separate the subheadings into two sections, such as "Selection of Studies" and "Characteristics of the Included Studies."

Line # 157……… This sentence should be placed at the beginning of paragraph, “Selection of studies”.

Line # 160, 165 and 166……… It is better to use "eligible". (Please follow this suggestion wherever it is applicable.)

Line # 162……… Source of the control participants?

Line # 167-168……… Please rephrase this sentence.

Table 1…….. Footnotes are incomplete. HB? PB?

Last column…….. Quality score?

Line # 175……… Please add the names of the genetic models for the audiences' convenience. Apply this recommendation to Table 2 as well by including a new column for the genetic model's name.

Line # 204-205………Please clearly indicate that the results are "results of a leave-one-out metanalysis" (also see my comments on Lines 53 and 141–142).

Study limitations

Line # 273…………….. Stability is an appropriate term. What about the reliability? Which analysis/results captured the reliability?

Line # 273…………….. Language bias or information bias?

Line # 275-283……………..Please rephrase the sentences and shorten it.

First, language bias/informational bias? Second, the confounding bias created by……. Third, not all studies conform to the HWE. So, what type of bias? Fourth, there could be additional factors that have not been included in this review.

Acknowledgements

Are you thanking your research team members or the participants and researchers in the studies included in your review? Please clarify.

---

## [Author Response · Author response to Decision Letter 2]

15 Jun 2023

Dear Editor and Reviewers,

Thank you very much for all your insightful comments and suggestions. They have been very helpful. We carefully modified the manuscript accordingly. All the revised portions of the manuscript are now marked in red (except in the clean version of the revised manuscript, also submitted). Please, see below for our responses to all the comments/suggestions.

Journal Requirements:

Response: Thank you for this comment. We have carefully reviewed reference list and ensureed that it is complete and correct.

Additional Editor Comments:

Title

Regarding my background comments, the phrase "an updated" has to be removed from the title.

Previous publications were not done by this team.

Response: Thank you for this comment. The phrase "an updated" has been removed from the title.

Abstract

Line #32………. “Obstructive” should not be capitalized

Response: Thank you for this comment. “Obstructive” has been capitalized.

Line #41……… Please include the names of genetic models that were created for overall population. I believe that overall population was assessed using five different genetic models such as allele. "Dominant, recessive, homozygous, and heterozygous models"

Response: Thank you for this comment. The names of genetic models that were created for overall population has been included.

Line # 47, 49……….. To spell out HB and PB

Response: Thank you for this comment. HB and PB have been spelled out.

Line # 50…………. Name the genetic models for AA in comparison to GG, AG/GG

Response: Thank you for this comment. We have named the genetic models for AA in comparison to GG, AG/GG.

Line # 53…………. The statement "a leave-one-out meta-analysis" should be applied.

Response: Thank you for this comment. The statement "a leave-one-out meta-analysis" has been applied.

Line # 56……… Take out the word “also”

Response: Thank you for this comment. The word “also” has been taken out.

TEXT

Line # 63……….. replacing "published studies reported" for "increasing evidence has demonstrated" 

Response: Thank you for this comment. We have replaced "published studies reported" for "increasing evidence has demonstrated".

Line # 68/ 69………. Please rephrase this sentence. The word "thus" was used twice

Response: Thank you for this comment. We have.rephrased this sentence.

Line # 72………. The wording "by Szily and coworkers or By Szily and associates" should have been used.

Response: Thank you for this comment. The wording "by Szily and coworkers" has been used.

Line # 72………. Replace the word "strong evidence" with lighter tone, like "published studies."

Response: Thank you for this comment. We have replaced "strong evidence" for "published studies".

Line # 78, 79, 80…… The authors could combine these two sentences into one and then cite [13] afterward.

Response: Thank you for this comment. We have combined these two sentences into one and then cited [13] afterward.

Line # 90………. “Hence” is a preferrable term.

Response: Thank you for this comment. We have replaced "Herein" for " Hence ".

Line # 92………. “Source of the control participants”

Response: Thank you for this comment. We have replaced "control source " for " source of the control participants".

Line # 114……… Please, remove “additionally”.

Response: Thank you for this comment. We have removed “additionally”.

Line # 133……… Please keep "homozygous" and "heterozygous" while removing "codominant."

Response: Thank you for this comment. We have removed “codominant”.

Line # 138……The word .”Otherwise” is preferrable to “By contrast”..

Response: Thank you for this comment. We have replaced "By contrast" for "Otherwise ".

Line # 141……. So as to differentiate it from statistical models (Random. Fixed models), it needs to be mentioned as "for all genetic models".

Response: Thank you for this comment. We have replaced "for all models " for " for all genetic models ".

Line # 141-142………… Are you referring to "a leave-one-out metanalysis" for the stability of the results.

Response: Thank you for this comment.Yes.

Line # 146……… Literature search and characteristics of included studies.

Please separate the subheadings into two sections, such as "Selection of Studies" and "Characteristics of the Included Studies."

Response: Thank you for this comment. We have separated the subheadings into "Selection of Studies" and "Characteristics of the Included Studies."

Line # 157……… This sentence should be placed at the beginning of paragraph, “Selection of studies”.

Response: Thank you for this comment. This sentence “The process of study selection was presented in Fig 1.” has been placed at the beginning of paragraph “Selection of studies”.

Line # 160, 165 and 166……… It is better to use "eligible". (Please follow this suggestion wherever it is applicable.)

Response: Thank you for this comment. We have replaced "enrolled " for " eligible ".

Line # 162……… Source of the control participants?

Response: Thank you for this comment. "Source of the control participants " has been used.

Line # 167-168……… Please rephrase this sentence.

Response: Thank you for this comment.We have rephrased this sentence.

Table 1…….. Footnotes are incomplete. HB? PB?

Last column…….. Quality score?

Response: Thank you for this comment.” PB,population-based; HB,hospital-based ” have been existed in footnotes. We have replaced " score " for " Quality score " in last column.

Line # 175……… Please add the names of the genetic models for the audiences' convenience. Apply this recommendation to Table 2 as well by including a new column for the genetic model's name.

Response: Thank you for this comment. We have added the names of the genetic models and included a new column for the genetic model's name in Table 2.

Line # 204-205………Please clearly indicate that the results are "results of a leave-one-out metanalysis" (also see my comments on Lines 53 and 141–142).

Response: Thank you for this comment. We have replaced "sensitivity analysis " for " eligible " leave-one-out meta-analysis".

Study limitations

Line # 273…………….. Stability is an appropriate term. What about the reliability? Which analysis/results captured the reliability?

Response: Thank you for this comment. “Reliability” has been removed.

Line # 273…………….. Language bias or information bias?

Line # 275-283……………..Please rephrase the sentences and shorten it.

First, language bias/informational bias? Second, the confounding bias created by……. Third, not all studies conform to the HWE. So, what type of bias? Fourth, there could be additional factors that have not been included in this review.

Response: Thank you for this comment. We have modified it as follows: First, language bias resulted from limition to publication in English and Chinese. Second, the confounding bias created by unavailable factors in most eligible studies like sex, age, body mass index, life style, AHI, comorbidities, other gene polymorphisms that may play important roles in the pathophysiology of OSA. Third, sampling bias resulted from enrolling a study unconforming to HWE law. Fourth, there could be additional factors that have not been included in this review.

Acknowledgements

Are you thanking your research team members or the participants and researchers in the studies included in your review? Please clarify.

Response: Thank you for this comment. We have modified it as follows: We appreciate the contribution of participants and researchers in the studies included in this review.

Thank you and best regards.

Yours sincerely,

Qiao ling Yao 

Corresponding author:Qiao-ling Yao,PhD

Department of Physiology, School of Basic Medical Sciences, Xinjiang Medical University, Urumqi, Xinjiang, China.

Email: 49165627@ qq.com

Mobile Number:15022917911

---

## [Editor Report · Decision Letter 3]

7 Aug 2023

Association of Tumor Necrosis Factor-α-308G/A polymorphism with the risk of obstructive sleep apnea: A meta-analysis of 14 case-control studies

PONE-D-22-04764R3

Dear Dr. Yao,

We’re pleased to inform you that your manuscript has been judged scientifically suitable for publication and will be formally accepted for publication once it meets all outstanding technical requirements.

Kind regards,

Cho Naing, MBBS, PhD, FRCP

Academic Editor

PLOS ONE

Additional Editor Comments (optional):

The authors have improved the manuscript, accordingly.

Thank you.
---

## [Editor Report · Acceptance letter]

11 Aug 2023

PONE-D-22-04764R3 

*Association of Tumor Necrosis Factor-α-308G/A polymorphism with the risk of obstructive sleep apnea: A meta-analysis of 14 case-control studies*

Dear Dr. Yao:

I'm pleased to inform you that your manuscript has been deemed suitable for publication in PLOS ONE. Congratulations! Your manuscript is now with our production department. 

Kind regards, 

on behalf of

Professor Cho Naing 

Academic Editor

PLOS ONE